# Household water and food insecurity negatively impacts self-reported physical and mental health in the Vietnamese Mekong Delta

Thuy Ngoc Vuong[1,2], Chinh Van Dang[2], Simon Toze[3], Paul Jagals[4], Danielle Gallegos[1,5‡]*, Michelle L. Gatton[6‡]

1 School of Exercise and Nutrition Sciences, Queensland University of Technology, Brisbane, QLD, Australia, 2 Institute of Public Health in Ho Chi Minh City, Ho Chi Minh City, Vietnam, 3 Urban Water Futures, Brisbane, QLD, Australia, 4 Children's Health and Environment Program, Child Health Research Centre, The University of Queensland, Brisbane, QLD, Australia, 5 Woolworths Centre for Childhood Nutrition Research, Faculty of Health, Queensland University of Technology (QUT), Brisbane, QLD, Australia, 6 School of Public Health and Social Work, Queensland University of Technology (QUT), Brisbane, QLD, Australia

‡ DG and MLG are joint senior authors on this work.
* danielle.gallegos@qut.edu.au

**Data Availability Statement:** The data underlying the results presented in the study are available from Vuong, Thuy Ngoc; (2021): Household food

## Abstract

### Introduction

Household food insecurity and inadequate water, sanitation, and hygiene (WASH) contribute to ill health. However, the interactions between household food insecurity, WASH and health have been rarely assessed concurrently. This study investigated compounded impacts of household food insecurity and WASH on self-reported physical and mental health of adults in the Vietnamese Mekong Delta.

### Materials and methods

This cross-sectional survey interviewed 552 households in one northern and one southern province of the Vietnamese Mekong Delta. The survey incorporated previously validated tools such as the Short Form 12-item Health Survey, Household Food Insecurity Assessment Scale, and the Access and Behavioural Outcome Indicators for Water, Sanitation, and Hygiene. Physical and mental health were quantified using the physical health composite score (PCS) and mental health composite score (MCS), respectively. These measures were the dependent variables of interest for this study.

### Results

Statistical analysis revealed that household food insecurity and using <50 litres of water per person per day (pppd) were independently associated with lower PCS (p<0.05), after adjusting for socio-economic confounders. Household food insecurity and lack of food availability, using <50 litres of water pppd, and the use of untreated drinking water were associated with lower MCS (p<0.05), with water usage being an effect modifier of the relationship between

insecurity data from the Vietnamese Mekong Delta. Queensland University of Technology. (Dataset) https://doi.org/10.25912/RDF_1636522923750

**Funding:** This study was funded by the Queensland University of Technology (QUT) Institute of Health and Biomedical Innovation (IHBI) and Institute of Public Health, Ho Chi Minh City.

**Competing interests:** This study had financial support from the Institute of Health and Biomedical Innovation, QUT and the Ho Chi Minh City Institute of Public Health to fund the research assistants who collected the data. Chinh Van Dang received salary from Ho Chi Minh City Institute of Public Health. Danielle Gallegos is currently supported by a grant from the Queensland Children's Hospital Foundation through a philanthropic donation from Woolworths. The Foundation and Woolworths did not provide any funding for this research, and neither organization had any role in the design, collection, analysis or interpretation of the data or in the writing of the manuscript. There are no restrictions on the sharing of data and materials.

household food insecurity and MCS. The results indicate that being food insecure and having limited potable quality water had a compounding effect on MCS, compared to being individually either food insecure or having limited water.

## Conclusion

This study is one of only a few that have established a link between potable water availability, food insecurity and poorer physical and mental health. The results also indicate a need to validate national data with fine-scale investigations in less populous regions to evaluate national initiatives with local populations that may be at higher risk. Adopting joint dual-action policies for interventions that simultaneously address water and food insecurity should result in larger improvements in health, particularly mental health, compared to targeting either food or water insecurity in isolation.

## Introduction

Ensuring adequate access to safe and healthy food as well as water, sanitation and hygiene (WASH) are priority global health and development goals due to their sustained and substantial contribution to reduced morbidity and mortality [1, 2]. It is estimated that globally more than 800 million people have limited access to adequate safe, nutritious food [3]; more than 785 million people do not have access to a basic drinking water service [4]; two billion people do not have basic sanitation facilities [5]; and one in four persons do not have access to a handwashing facility with soap and water [2].

Inadequate WASH and food insecurity has been linked to poorer physical health [6]. Increasingly however, using untreated drinking water in low- and middle-income countries (LMICS) has been found to impact mental health by increasing depression and anxiety [7, 8]. There is also evidence that individuals living in food-insecure households are at higher risk of compromised mental health [9–11]. Consequently, food insecurity and inadequate WASH may interact to compound the detrimental impact on physical and mental health. However, our understanding of the interaction between inadequate WASH and food insecurity on people's health, particularly their mental health, is limited [12]. The interactions between food availability and access, WASH and health have rarely been assessed concurrently [6] especially not in the VMD.

The Vietnamese Mekong Delta (VMD), where this research was conducted, is a rural area and leading producer and exporter of agricultural products particularly rice, fruits, and aquaculture (fish and shrimp) [13]. This region is experiencing increasing compromised food and potable water security due to climate change, impacts of technical agricultural and economic developments, and poor infrastructure [14–16]. The VMD is experiencing prolonged periods of drought interspersed with flooding events. Saltwater intrusion into fresh groundwater due to over-extraction has also led to shortages of fresh water for food production and daily household use especially during the dry season [16]. Scarcity of clean water is further worsened by a shortage of adequate wastewater treatment facilities in the region, leading to untreated wastewater from both domestic and industrial sources being directly discharged into rivers and canals [17]. Food and water supplies have been further compromised by water pollution that results from overuse of agricultural chemicals and fertilizers in farming and industrial activities, soil salinisation and high natural concentrations of aluminium in soils in the region [14,

16, 18]. Food insecurity is also amplified by population migration to urban areas, thus reducing the labour force for agricultural activities and limiting production and subsequently incomes [16].

In terms of sanitation, Vietnam has improved the national coverage of basic sanitation facilities for the population, from 52% in 2000 to 84% in 2017 [19]. However, data on basic sanitation facilities at a finer scale shows that access to basic sanitation facilities in rural areas (78%) [19], ethnic minority groups (61%) as well as those living with poverty (41%), remains lower when compared to their urban (94%) or wealthier counterparts (98%) [20]. Over half (55%) of households in the VMD use unimproved sanitation facilities and the most common sanitation facility type are hanging toilets consisting of simple toilet structures suspended over a body of water that release excreta directly into water bodies [21]. Only 13% of the rural population in Vietnam have been documented to wash their hands with soap at critical moments and this rate is even lower among ethnic minority groups [19]. There is little information on hand-washing with soap, particularly in the VMD. The lack of access to clean and safe water, coupled with poor sanitation and hygiene practices may contribute to high rates of diarrhea, pneumonia, and parasitic infections resulting in subsequent undernutrition, especially in children [22].

This study therefore aimed to determine the relationships between food and the availability of sufficient water of appropriate quality for potable consumption purposes (termed "domestic potable water security" in this study) along with inadequate WASH on the surveyed physical and mental health of rural households in one northern and one southern region of the VMD. The working hypothesis was that the interactions between food insecurity and inadequate WASH influenced the physical and mental health of primary food preparers (PFP) and by inference, their households, independent of socio-economic factors.

## Materials and methods

### Study area and population

The VMD has a population of more than 17 million people with one major city, Can Tho, and 12 provinces [18]. The VMD can be divided into three agricultural regions: north (rice and freshwater aquaculture), central (rice, freshwater and marine aquaculture), and south (marine aquaculture as a result of limited fresh water supply coupled with increased saline intrusions) [18]. This study was conducted in 2018 in two rural provinces, one located in the northern region, and the other in the southern region of the VMD. Primary food preparers from 552 randomly households, 226 in each province participated in the study. All selected households participated in the study and provided a full complement of data.

### Ethics approval

Ethics approval for this study was obtained from the Queensland University of Technology Human Research Ethics Committee, Australia (QUT UHREC 1700000907) and the Institute of Public Health of Ho Chi Minh City, Vietnam (122/TB-VYTCC). Households who agreed to take part in the study after being provided the information about the study provided written consent. Participants who were unable to read and write were informed verbally by the interviewer and provided verbal consent in the presence of a member of the communal health staff as a witness. Each household was paid cash compensation for their participation of VND 50,000 (approximately $2 USD).

**Inclusivity in global research.**   Additional information regarding the ethical, cultural, and scientific considerations specific to inclusivity in global research is included in the S1 Checklist.

## Survey and data collection

The survey was developed to be collected using electronic tablets and had two components: a structured questionnaire and an observation checklist, both administered by a trained interviewer. The questionnaire was used to collect data from the PFP on socio-economic characteristics and agricultural practices of the household, HFI, household food availability in the previous month, diet diversity, WASH, and self-reported physical and mental health status. The observation checklist was used to collect data on water storage, toilet type and sanitation practices. The survey instrument, observation checklist and survey data underlying the results presented in the study are available from Vuong, Thuy Ngoc; (2021): Household food insecurity data from the Vietnamese Mekong Delta. Queensland University of Technology. (Dataset) https://doi.org/10.25912/RDF_1636522923750. The questionnaire and observation checklist were initially piloted in 50 households to ensure face validity and repeated in 20 households to test reliability.

**Self-reported health status.** Self-reported health status is a commonly used measure in which respondents rate their own health, as excellent, very good, good, fair, or poor [23]. The Short Form-12 Health Survey (SF-12) [23] was used to assess health status. The SF-12 has been previously validated in LMICS and used in Vietnam [24–26] with a recall period of one month. The Cronbach's α was excellent in this study at 0.94. This instrument covers eight health domains which were scored separately: (i) physical functioning (PF), (ii) physical role (PR), (iii) bodily pain (BP), (iv) general health GH, (v) vitality (VT), (vi) emotional role (ER), (vii) social functioning (SF), and (viii) mental health (MH) [23]. In addition, the physical health composite score (PCS) and mental health composite score (MCS) were calculated by aggregating domain scores as per the recommended procedure [23]. Only the PCS and MCS are reported here. These scores were standardized to a mean of 50 and the standard deviation of 10 according to the US population, with higher scores indicating better health [23].

**Socio-economic data, agricultural practices, and self-reported shocks in the previous year.** Data were collected on sex, ethnicity, education, occupation, family structure (family size, household head, marital status), household income and expenditure, debts, household assets, land ownership, farming activities, and produce storage for domestic use.

As part of the questionnaire, a Shock Scale between 0 and 9 was created by counting of how many of the following nine nominated events occurred in the previous year, with each event that household experienced being scored 1 [27]: 1) crop loss due to extreme weather; 2) crop loss due to other reason; 3) death of main income earner; 4) death of other income earners; 5) disease or injury of any family member that reduced income; 6) loss of employment of any family member; 7) loss of cattle (due to disease, injury or other); 8) damage to house or any productive assets (theft, fire, river erosion, cyclone etc.); 9) and business failure [27]. Higher scores indicate higher levels of shock.

**Household food insecurity, diet diversity, and food availability.** The Household Food Insecurity Assessment Scale (HFIAS) (Cronbach's α = 0.97, excellent), Household Diet Diversity Score (HDDS) (Cronbach's α = 0.65, moderate), and Home Food Environment Survey (HFES) (Cronbach's α = 0.90, good) [28–30] were used to evaluate food insecurity, diet diversity, and food availability respectively. HFIAS scores ranged from zero to 27, with higher scores indicating more severe food insecurity. HFIAS scores were also dichotomized into food secure (0–1 score) and food insecure (2+ scores) for further analyses [28]. Total scores of diet diversity and availability ranged from 0–12, with higher scores indicating higher diet diversity and food availability.

**Potable water security and Water, Sanitation, and Hygiene (WASH).** Potable Water Security and WASH conditions were measured through direct questioning and observation.

Direct questions on WASH included access to and use of different sources of water for domestic potable water uses, proxies of water quality such as drinking water treatment and water storage, the estimated volume of water used per person per day (pppd) for domestic potable water uses, types of sanitation facilities, and sharing of those facilities, and hand-washing practices at key moments. The study only considered water used within the house and specifically for drinking and cooking purposes. Other uses within the home such as bathing and washing were not considered except as part of the volume used per person per day for all domestic purposes.

In addition, certain WASH components were observed by trained interviewers using the observation checklist. The observations included condition of household water containers (lid, spigot, and presence of mosquito larvae), sanitation facilities (type, cleanliness of pathway to sanitation facility and inside facilities), and hand-washing facilities (presence of soap and water). The questions and observations about drinking water, sanitation, and proxies for hygiene practices were compiled from several tools that have been used internationally in a range of contexts [31, 32].

Several key variables were created from the data. The source and treatment of the drinking water after collection were combined to produce a 'drinking water treatment' variable with three categories: 'point-of-use treated' (POUT) (boiled by the participant prior to use) when households use POUT water for drinking in both the rainy and dry seasons, 'out-side treated' when bottled or piped (but not boiled) drinking water was used in both the dry and rainy seasons, and 'untreated' when households used untreated water from environmental sources such as river/canal water, rainwater and tube-well or drilled well water (but not boiled) for drinking in either the rainy or dry seasons (or both).

Quantity of water used for all potable domestic uses pppd was calculated by dividing the total quantity of water used per household by the total number of people in the house; this was then dichotomized into 50+ litres (as of minimum water quantity needed for each individual domestic use per day) and < 50 litres [33]. The volume of domestic potable water use pppd was used as an indication of the amount of water (regardless of source) available for drinking and cooking uses. The condition of outside water storage containers for households were also observed directly, noting whether their water containers had a lid, a spigot and no mosquito larvae, thus representing proper water storage [34].

Sanitation facilities were categorised as: 'basic' when households use flush or pour flush toilets, pit toilets, or composting toilets which they do not share with other households; 'limited' when households use flush or pour flush toilets, pit toilets, or composting toilets which they share with other households; 'unimproved' when households use hanging toilets; and 'no facilities' when households do not have a dedicated toilet [32]. Sanitation facilities were then categorized into improved (basic and limited), unimproved, or no facility. Handwashing facilities were categorized as 'basic' when households had handwashing facilities with water and soap available at the time of observation, and 'limited' when households had handwashing facilities but lacked either water or soap or both [35]. Finally, PFP were noted as having "handwashing with soap" knowledge if they knew all five critical moments of handwashing with soap (after defecation, after cleaning or toileting a child, before preparing food, before feeding a child, and before eating) [34]. Reliability of the WASH components included in the observation checklist was assessed using Intra-correlation Coefficient (ICC) being rated as excellent for quantity of water used pppd (ICC = 0.93; 95% CI: 0.82–0.97), moderate for water storage (ICC = 0.61; 95% CI: 0.25–0.80) and handwashing facilities (ICC = 0.56; 95% CI: 0.16–0.83), and good for handwashing with soap (ICC = 0.79; 95% CI: 0.46–0.92).

## Statistical analysis

Data analyses were conducted using Stata 15.0 [36]. In the study area the large majority of households were Kinh in both provinces, with 99% of Khmer households located in the southern province. Consequently, three population groups were created for analysis: northern province Kinh, southern province Kinh, and Khmer. Bivariate associations between the physical health composite score (PCS) and the mental health composite score (MCS) and population group were tested using ANOVA coupled with Bonferroni post-hoc testing. In multivariate analysis on factors associated with PCS and MCS, each of the PCS and MCS associations was evaluated using general linear models at the first stage. Then, each model was adjusted for the other health outcomes as there is a correlation between PCS and MCS (r = 0.2; p< 0.01). For example, the model examining risk factors of PCS was adjusted for MCS. Independent variables with significant collinearity (VIF >5) were excluded.

## Results

The socio-economic characteristics of the participants were compared to those reported for greater Vietnam and the entire VMD in the Vietnam Household Living Standard Survey published in 2018 (Table 1) [37]. The surveyed households in this current study were more disadvantaged when compared to the overall VMD and Vietnamese population, with a higher proportion of PFP never having gone to school, or with a primary school education, and a lower average monthly household income. Females were over-represented in those surveyed (81.5% of sample), as expected due to surveying PFP. All those who agreed to participate, completed all parts of the relevant surveys with no withdrawals at any stage of the study.

Just over one-third (34.4%, 190/552) of surveyed households were food insecure in the last month. On average, each household had 10 (± 1.8, range: 3–12) food groups available in the past month and consumed 6 (± 1.7, range: 2–11) different food groups on the previous day.

Overall, 32.8% of households used (purchased) bottled water as their primary source of drinking and cooking water, followed by rainwater (26.5%), piped water (22.1%) and creek/lake/stream/pond water (15.0%) (S1 Table). However, there were considerable differences in drinking water sources between the northern and southern provinces, most notably the dominance of rainwater in the southern province and the use of creek/lake/stream/pond water in the northern province but not the southern province (S1 Table). The majority (65.8%) of households undertook some sort of treatment of the water prior to consumption, with boiling being the predominant treatment method.

Differences in WASH variables were compared between the three population groups (Table 2). Kinh households in the southern province were significantly more likely to use untreated drinking water compared to either their northern province Kinh counterparts, or the Khmer (p<0.0001). For the remainder of the WASH components, there were no significant differences between Kinh households in the northern and southern provinces. In contrast, Khmer households were more likely to not own sanitation facilities (p = 0.003) and have limited handwashing facilities (p = 0.044) and knowledge of handwashing with soap (p = 0.001), compared to their southern province Kinh counterparts. There was also a trend for Khmer to be more likely to use <50 litres of water pppd (p = 0.058) for domestic potable uses and have improper water storage (p = 0.07), however these differences failed to reach statistical significance.

Regarding health status, the overall mean PCS and MCS for primary food preparers were 42.9 (±9.0) and 47.3 (±10.1), respectively. Primary food preparers belonging to the Khmer minority group had a mean MCS of 44.0 (±11.7) which was significantly lower than for PFP in

**Table 1. Selected socio-economic characteristics, water and sanitation, of the study sample compared with the national and regional population.**

| Characteristics | PFP in current study % | VMD population % [α] | Vietnamese population % [α] |
|---|---|---|---|
| **Ethnicity** | | | |
| Kinh (majority ethnic group) | 80.0 | 92.0 | 86.0 [a] |
| Khmer | 20.0 | 6.4 | 1.5 [a] |
| Others (Muong, Hoa, Cham, H'mong…) | 0.0 | 1.6 | 12.5 [a] |
| **Sex** | | | |
| Male | 18.5 | 49.0 [b] | 49.0 |
| Female | 81.5 | 51.0 [b] | 51.0 |
| **Marital status** | | | |
| Never married | 5.1 | NA | 24.6 |
| Married | 82.6 | NA | 65.8 |
| Divorced/widowed/separated | 12.3 | NA | 9.6 |
| **Educational level** | | | |
| Never gone to school | 13.2 | 6.3 | 5.2 |
| Primary school (grades 1 to 5) | 42.2 | 31.2 | 31.9 |
| Secondary school (grades 6 to 9) | 29.7 | 19.7 | 28.3 |
| High school and above | 14.9 | 19.4 | 34.6 |
| **Main drinking water sources** | | | |
| Piped water | 22.1 | 44.4 | 43.4 |
| Well water | 4.0 | 23.7 | 36.2 |
| Rainwater | 26.1 | 19.1 | 9.4 |
| Purchased water | 32.8 | 0.1 [c] | 0.2 [c] |
| River/canal water | 15.0 | NA | NA |
| Filtered spring water | 0.0 | 0.2 | 4.7 |
| Other | 0.0 | 12.2 | 5.8 |
| **Sanitation facilities** | | | |
| Improved toilets | 72.6 | 77.6 [d] | 92.0 [d] |
| Hanging toilets | 20.1 | 20.9 | 4.2 |
| Other | 7.3 | 1.5 | 3.8 |
| **Average monthly household income (VND'000)** | 3,004 | 3,585 | 3,873 |
| **Average monthly food expenditure (VND'000)** | 2,200 | 995 [e] | 1,118 [e] |
| **Average household size (persons)** | 4.0 | 3.6 | 3.7 |

[α] Most data were extracted from the 2018 Vietnam Household Living Standard Survey. Data were from other sources are indicated as indicated below.

a: Data were sourced from https://worldpopulationreview.com/countries/vietnam-population/ (Accessed on 23 June 2020).

b: Source: Cosslett, T. L., & Cosslett, P. D. (2014). Water Resources and Food Security in the Vietnam Mekong Delta (Vol. 44): Springer International Publishing.

c: Purchased water is bottled water of unknown quality purchased for drinking and cooking purposes.

d: Flush toilet with septic tank, sewage pipes and double vault compost latrines.

e: Eating, drinking & smoking.

NA = Not available.

Kinh households in the northern (47.8±9.0) or southern (48.4±10.2) provinces (p<0.0001). There was no significant difference in mean PCS between population groups (p = 0.3) (Fig 1).

## PCS and associated factors

HFI and quantity of water used (pppd) were separately associated with lower PCS after adjustment for other variables (Table 3). The estimated marginal mean PCS of PFP who lived in food insecure households was 2.94 points lower than those who lived in food secure households (95% CI: 1.31–4.57; p<0.0001). The estimated marginal mean PCS of PFP who lived in

**Table 2. Differences in WASH components by population groups.**

| | Total | | NP Kinh | SP Kinh | Khmer | RRR (95% CI) | RRR (95% CI) |
|---|---|---|---|---|---|---|---|
| | n | % | % | % (ref) | % | NP Kinh vs. SP Kinh | Khmer vs. SP Kinh |
| **Drinking water treatment[a]** | | | | | | | |
| Point of used treated (POUT) | 340 | 61.6 | 66.2 | 53.0 | 63.3 | ref | ref |
| Out-side treated | 148 | 26.8 | 32.0 | 20.2 | 23.9 | 1.3 (0.8–2.1) | 1.0 (0.5–1.8) |
| Untreated | 64 | 11.6 | 1.8 | 26.8 | 12.8 | 0.1 (0.0–0.1) ** | 0.4 (0.2–0.8) ** |
| **Quantity of water used pppd** | | | | | | | |
| 50+ litres | 436 | 79.0 | 82.2 | 79.8 | 69.7 | ref | Ref |
| <50 litres | 116 | 21.0 | 17.8 | 20.2 | 30.3 | 0.9 (0.5–1.4) | 1.7 (1.0–3.0) |
| **Water storage[b]** | | | | | | | |
| Proper | 196 | 35.5 | 40.0 | 35.1 | 24.8 | ref | Ref |
| Improper | 356 | 64.5 | 60.0 | 64.9 | 75.2 | 0.8 (0.5–1.2) | 1.6 (1.0–2.8) |
| **Sanitation facilities[c]** | | | | | | | |
| Improved | 401 | 72.6 | 77.8 | 71.4 | 61.5 | ref | Ref |
| Unimproved | 111 | 20.1 | 18.2 | 22.6 | 21.1 | 0.7 (0.5–1.2) | 1.1 (0.6–2.0) |
| No facility | 40 | 7.3 | 4.0 | 5.9 | 17.4 | 0.6 (0.3–1.5) | 3.4 (1.5–7.7)** |
| **Handwashing facility[d]** | | | | | | | |
| Basic | 273 | 49.5 | 53.8 | 50.0 | 37.6 | ref | Ref |
| Limited | 279 | 50.5 | 46.2 | 50.0 | 62.4 | 0.9 (0.6–1.3) | 1.7 (1.0–2.7)* |
| **Handwashing with soap knowledge[e]** | | | | | | | |
| Yes | 112 | 20.3 | 25.1 | 21.4 | 6.4 | ref | Ref |
| No | 440 | 79.7 | 74.9 | 78.6 | 93.6 | 0.8 (0.5–1.3) | 4.0 (1.7–9.3)** |

NP, Northern Province; SP, Southern Province; ref, reference; RRR, relative risk ratio; CI, confidence interval; POUT (point of use treated).

*significant at p < .05

**significant at p < .01.

[a] POUT: Households use boiled water for drinking in both rainy and dry seasons; Out-side treated: Households use unboiled out-side treated water for drinking in both rainy and dry seasons; Untreated: Households use unboiled, untreated water for drinking in rainy or dry season or in both seasons.

[b] Proper: Households have an outside water container which had a lid/cover, a spigot and no larvae; Improper: Households have an outside water container which don't comply with at least one criterion.

[c] Basic: Households use flush or pour flush toilets, pit toilets, composting toilets and do not share with other households; Limited: Households use flush or pour flush toilets, pit toilets, composting toilets and share with other households; Unimproved: Households use hanging toilets.

[d] There were no households which did not have a handwashing facility in the sample; Basic: Households had handwashing facilities with water and soap available at the time of observation; Limited: Households have handwashing facilities but lacked either water or soap or both.

[e] Yes: Primary food preparers know all the five critical moments of handwashing with soap including after defecation, after cleaning a child or after toilet, before preparing food, before feeding a child, and before eating.

households using less than 50 litres pppd was 1.78 points lower than those who lived in households using 50 litres and above for domestic potable purposes after adjusting for other factors (95% CI: 0.14–3.42; p = 0.034).

## MCS and associated factors

The use of untreated water for potable uses, HFI, insufficient food availability and using <50 litres of water pppd for domestic purposes were all associated with lower MCS (Table 4). There was a significant interaction between HFI and quantity of water used pppd (p = 0.002). For households that were food secure, the quantity of water used domestically did not significantly influence the MCS (p>0.05). For food insecure households that used 50+ litres of water pppd for domestic uses, the MCS was predicted to be 2.99 points lower (95% CI: 1.07–4.92; p = 0.002) than the MCS in food secure households using 50+ litres of water pppd. When

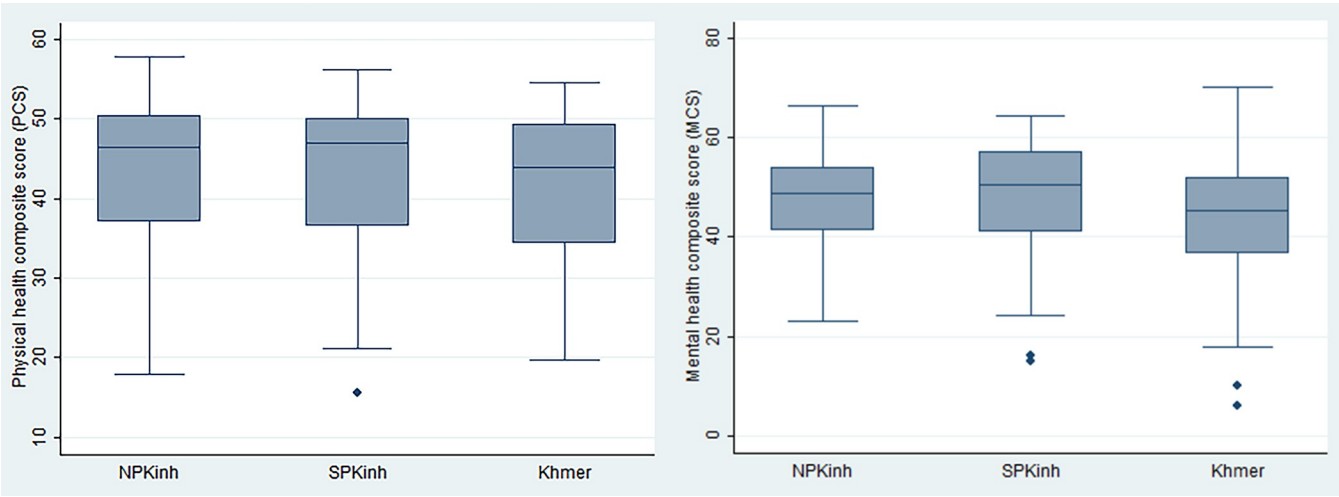

**Fig 1. PCS and MCS of primary food preparers by population group.**

the food insecure households used <50 litres of water pppd, then the MCS was predicted to decrease a further 5.95 points (95% CI: 2.18–9.73; p = 0.002).

## Discussion

The results from this study indicate that household food insecurity and limited quantity of potable domestic water use or the use of untreated (unboiled) drinking water were associated

**Table 3. Factors associated with PCS in multivariable general linear model.**

| | Estimated Marginal Mean PCS (95% CI) | p value |
|---|---|---|
| **Age of primary food preparers** (ref: 18–44 years) | | |
| 45–65 years | -4.53 (-5.91, -3.14,) | **<0.0001** |
| ≥ 65 years | -8.38 (-10.62, -6.14,) | **<0.0001** |
| **Family structure** (ref: Typical family) | | |
| Only couple | -2.82 (-5.27, -0.37,) | **0.024** |
| Widowed/single living with others | -2.83 (-5.13, -0.53) | **0.016** |
| Widowed/single alone | -4.70 (-8.08, -1.32) | **0.006** |
| **Engaging in farming** (ref: No) | | |
| Yes | 2.05 (0.51, 3.59) | **0.009** |
| **Quantity of water pppd** (ref: 50+ litres) | | |
| < 50 litres | -1.78 (-3.42, -0.14) | **0.034** |
| **Household food insecurity** (ref: Food secure) | | |
| Food insecure | -2.94 (-4.57, -1.31) | **<0.0001** |
| **Shock score** | -1.24 (-2.08, -0.42) | **0.003** |

PCS, physical composite health score; CI, confidence interval; ref, reference; MCS, mental composite score; pppd, per person per day.

Only factors with significant differences with p<0.05 or 0.01 are reported.

Variables included in the PCS model: HFI, age, occupation, income sources, income sufficiency, monthly household expenditure, saving, ability to access markets, food source, engaging in farming, number of people in the household, shock score, MCS, household diet diversity and food availability scores, drinking water treatment, quantity of water used pppd, water storage, sanitation facilities, handwashing facilities, and handwashing with soap knowledge.

**Table 4. Factors associated with MCS in multivariable general linear model.**

| Variables | Difference in Estimated Marginal Mean MCS (95% CI) | p values |
|---|---|---|
| **Monthly household income (VND)** (ref: $\geq 6$ million) | | |
| 3–5 million | -3.02 (-4.74, -1.30) | **0.001** |
| < 3 million | -4.19 (-6.11, -2.27) | **<0.0001** |
| **Drinking water treatment** (ref: Point of use treated) | | |
| Out-side treated | -1.59 (-3.21, 0.02) | 0.053 |
| Untreated | -3.26 (-6.11, -0.42) | **0.025** |
| **Household food insecurity** (ref: Food secure) | | |
| Food insecure | -2.99 (-4.92, -1.07) | **0.002** |
| **Food availability score** | 0.95 (0.41, 1.49) | **0.001** |
| **HFI*quantity of water used pppd** | -5.95 (-9.73, -2.18) | **0.002** |

MCS, mental composite score; CI, confidence interval; VND, Vietnamese dong; Ref, reference; HFI, household food insecurity; pppd, per person per day.

Only factors with significant differences with p<0.05 or 0.01 are reported.

Variables included in the MCS model: population group, monthly household income, HFI, PCS, household diet diversity and food availability scores, drinking water treatment, quantity of water used pppd.

with poorer physical and mental health, as experienced by the primary food preparer using self-reported measures. Specifically, the study found that HFI and using <50 litres of water for domestic use pppd were independently associated with lower PCS. With regards to mental health, HFI, lack of food availability, using <50 litres of water pppd, and the use of untreated drinking water lowered the MCS of adult primary food preparers living in rural areas of the VMD. Other WASH conditions such as sanitation and hygienic factors were not significantly associated with PCS or MCS of those surveyed in the VMD.

At the time of this study, it was one of only a few published studies demonstrating a negative interaction between HFI, and access to appropriate quantities of potable quality water used pppd and the mental health of adults living in a lower-middle income country. The results from this study undertaken in Vietnam adds to the recent data from other lower-middle income countries [38, 39]. This finding is important for global health policy, which currently predominantly considers water and food as independent factors which undermine health status [6]. The finding of this study therefore supports the need for a connected systems science approach to addressing food and water insecurity concurrently to improve health status for people since mitigating either food or water insecurity independently could be insufficient [6].

Water security has been defined as "the capacity of a population to safeguard sustainable access to adequate quantities of acceptable quality water for sustaining livelihoods, human well-being, and socio-economic development, for ensuring protection against water-borne pollution and water-related disasters, and for preserving ecosystems in a climate of peace and political stability" [40]. In this study we focused on potable uses of water in the home only with consideration of the four dimensions of water security: availability (safe water is available for use that is of sufficient volume to meet household needs), access (having economic, physical and social access to a safe drinking water source), utilisation (water quality or safety) and stability (stable over time irrespective of seasonal and civic changes). In this way, potable water security can be characterised similarly to household food security, indicating that potable water security is an under-conceptualised and probably under-researched dimension of food

security, and that pathways linking water insecurity to poor health are likely similar to those linked to household food insecurity [6]. In the VMD, many households rely on water sources that can be compromised. In general, the quality of the three main water sources (surface water, groundwater, and rainwater) in the VMD is increasingly threatened by a number of factors. The surface water and groundwater are largely contaminated by overuse of agricultural chemicals and fertilizers in farming and industrial activities, salinity intrusion and aluminium soils [17]. The threat to the quality of these water sources is further worsened because there is a shortage of wastewater treatment plants in the region [17], meaning wastewater from both domestic use and industrial parks is directly discharged into rivers and canals. Thus, surface water and groundwater are largely polluted. Over 50% of the households surveyed in our study boiled water prior to consumption irrespective of the original source of the water. It was for this reason that we captured this behaviour in the final classification of drinking water source. We acknowledge that behavioural adherence to correctly treat water by boiling is not always correctly adhered to. However, we saw boiling water before consumption as less of a problem than using a classification based on the original water source, which then may or may not be subsequently boiled. In addition, bottled water was not considered as treated potable water in this study as bottled water is purchased in the VMD at relatively low cost but tends to have poor water quality due to inadequate management [41]. Specifically, authorities that are responsible for managing quality of bottled water lack the resources to adequately supervise and control companies which produce this water source [41].

Both potable water insecurity and household food insecurity may be linked to health via biological, nutritional, and psychosocial pathways [40]. For example, household food insecurity is associated with inadequate energy intake and inadequate consumption of essential nutrients and vitamins [42], while water insecurity may force people to use poorer quality water, increasing the risk of diarrhoea from a range of infectious and parasitic infections and preventing adequate absorption of energy and micronutrients [22]. Intestinal parasitic infections can also affect people's appetite which additionally contributes to undernutrition [43]. Additionally, people living in food insecure households that also have limited access to potable water may experience feelings of uncertainty or stress over how long their food and water of appropriate quality will last, which can generate other negative feelings such as deprivation, anxiety, shame, and helplessness [44, 45].

This study found that the mental health of people living in food insecure households that used less than 50 litres pppd was worse than that of households who were food insecure but were able to use 50 litres and above pppd of appropriate quality water. This finding extends prior work which showed that food and water insecurity independently affects mental health [46, 47]. While it has been reported that food insecurity is independently linked to emotional distress, anxiety, depression and other mental health outcomes [48, 49], this study suggests that mental health may also be further compromised by the interactive effect of food insecurity and other 'insecurities' (such as potable water insecurity). There is increasing recognition of the importance of the integrated assessment of multiple insecurities and their interactions to identify the common determinants leading to ill health in order to design double and triple-duty actions to improve human health [38, 50, 51]. Thus, future studies that explore the cumulative and interactive effects of stressors from water, food and other insecurities (energy and income losses) will be important for understanding the influence shared drivers have toward efficient multiple action approaches to improving health.

This study also found that using untreated potable water, assessed as the combination of an unimproved water source and not boiling the water for consumption after collection, was associated with increasingly negative mental health. This finding is also novel for its assessment of measures to assess water quality in relation to health given the lack of research on household

water insecurity [6]. Collective findings of this study therefore suggested that the lack of both quantity and quality of water were associated with a decrease in mental health. It should also be noted that although boiling (the most common household water treatment practice in VMD) can disinfect water, it cannot guarantee the removal of chemical contaminants such as arsenic or mercury, which are commonly found in tube-wells or drilled wells in VMD [52]. Therefore, household water treatment models that address not only pathogens and parasites but also chemicals in VMD need to be prioritised to ensure water security and health for people. Additionally, improving access to safe water can have a flow-on effect by improving the opportunities for households to improve food security through ensuring food safety (such as using clean and safe water for food preparation and cooking) or through improved health of the primary food providers [12].

Although the results of this study did not show an accumulative effect of HFI and potable water insecurity on physical health, each of these insecurities was independently associated with lower physical health, which is consistent with previous research [6, 53]. This study also extends beyond prior work for its findings which showed that both HFI and potable water insecurity impacted collectively on both physical health and mental health and on the shocks that household experienced such as crop loss due of extreme weather, loss of employment, and the death of household income earners. This finding demonstrates that physical health will be more likely to be undermined by multiple insecurities such as food insecurity, water insecurity, and personal wellbeing insecurities.

Collective findings of this study demonstrate that both physical and mental health are more likely to be undermined by multiple insecurities particularly potable water, food, and wellbeing insecurity, contributing to a broader understanding of shared drivers of ill-health. This study also employed reliable tools which were carefully piloted for reliability in the VMD context, along with robust statistical analyses to assess the relationships between outcomes and associated factors. The results also highlight the need to validate national data of a country with fine scale investigations in regions away from major population centres, These investigations are needed to evaluate national improvement initiatives and with local populations that may be at higher risk. This can assist governments and aid agencies to further refine assistance that ensures these at-risk populations also benefit from national improvements in food and water access.

However, the study design of this and other previous studies does not make it possible to determine an overall direct cause-and-effect relationship between household food and potable water insecurity and health status. Therefore, future studies should be designed to better ascertain these links, for example, more in-depth epidemiological studies of health associated with both HFI and potable water insecurity. In addition, this study did not examine the extent to which seasonal food and potable water insecurities impact on health, especially mental health. Future studies to fill these gaps and to contribute to the literature are warranted. In addition, the use of self-reporting can create the potential for social desirability bias; however, this was mitigated to some extent by independent observation of WASH components. The use of an interviewer-administered survey may have raised expectations about potential access to resources or conversely limited reporting of hardship to minimise shame. This bias was partially ameliorated by clearly explaining to participants that the findings of the research only served the aims of the study and did not relate to local or national welfare programs.

## Conclusions

Our findings demonstrate that potable household water and food insecurity may be independently or jointly associated with poorer physical and mental health in the VMD. This is one of

the few studies demonstrating links between potable water and food insecurity with issues of physical and mental health. The results also suggest that the simultaneous development and implementation of joint policies and programs that examine potential problems at a local, downsized scale to resolve these two conditions are warranted to improve health at a finer scale than national level. For instance, joint dual-action interventions could address food and potable water insecurity more effectively for improving the population health since food and water insecurity may have shared common drivers and solutions such as socio-economic (poverty) and environmental factors (in particular increasing impacts of climate change). This study also supports a broader global focus on achieving multiple goals and targets in the Sustainable Development Goals by integrated policies and interventions to offset limited global resources.

## Supporting information

**S1 Checklist. Checklist for global inclusivity.**
(DOCX)

**S1 Table. Details of drinking water source (DWS) by population group (n = 552).**
(DOCX)

## Acknowledgments

We acknowledge support from local commune staff and participants in the study areas. We are also grateful to the staff of the Institute of Public Health (IPH) for their assistance in data collection.

## Author Contributions

**Conceptualization:** Thuy Ngoc Vuong, Simon Toze, Paul Jagals, Danielle Gallegos.

**Data curation:** Thuy Ngoc Vuong.

**Formal analysis:** Thuy Ngoc Vuong, Danielle Gallegos, Michelle L. Gatton.

**Funding acquisition:** Chinh Van Dang.

**Investigation:** Thuy Ngoc Vuong.

**Methodology:** Thuy Ngoc Vuong, Simon Toze, Danielle Gallegos, Michelle L. Gatton.

**Project administration:** Thuy Ngoc Vuong, Chinh Van Dang, Danielle Gallegos, Michelle L. Gatton.

**Resources:** Chinh Van Dang.

**Supervision:** Chinh Van Dang, Simon Toze, Paul Jagals, Danielle Gallegos, Michelle L. Gatton.

**Writing – original draft:** Thuy Ngoc Vuong.

**Writing – review & editing:** Thuy Ngoc Vuong, Chinh Van Dang, Simon Toze, Paul Jagals, Danielle Gallegos, Michelle L. Gatton.

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
