## [Decision Letter · Decision Letter 0]

4 Feb 2022

PONE-D-21-39513Household water and food insecurity negatively impacts self-reported physical and mental health in the Vietnamese Mekong DeltaPLOS ONE

Dear Dr. Gallegos,

Thank you for submitting your manuscript to PLOS ONE. After careful consideration, we feel that it has merit but does not fully meet PLOS ONE’s publication criteria as it currently stands. Therefore, we invite you to submit a revised version of the manuscript that addresses the points raised during the review process. You will see that both reviewers have questions about your conceptualisation of water security.   For reviewer 2 this is a major concern, but I think reviewer 1's comments which they say are "for another paper" may help you address these.  

We look forward to receiving your revised manuscript.

Kind regards,

Alison Parker

Academic Editor

PLOS ONE

Journal Requirements:

1. When submitting your revision, we need you to address these additional requirements. Please ensure that your manuscript meets PLOS ONE's style requirements, including those for file naming. The PLOS ONE style templates can be found at https://journals.plos.org/plosone/s/file?id=wjVg/PLOSOne_formatting_sample_main_body.pdf and https://journals.plos.org/plosone/s/file?id=ba62/PLOSOne_formatting_sample_title_authors_affiliations.pdf 2.  Please include a complete copy of PLOS’ questionnaire on inclusivity in global research in your revised manuscript. Our policy for research in this area aims to improve transparency in the reporting of research performed outside of researchers’ own country or community. The policy applies to researchers who have travelled to a different country to conduct research, research with Indigenous populations or their lands, and research on cultural artefacts. The questionnaire can also be requested at the journal’s discretion for any other submissions, even if these conditions are not met.  Please find more information on the policy and a link to download a blank copy of the questionnaire here: https://journals.plos.org/plosone/s/best-practices-in-research-reporting. Please upload a completed version of your questionnaire as Supporting Information when you resubmit your manuscript. 3. We note that the grant information you provided in the ‘Funding Information’ and ‘Financial Disclosure’ sections do not match.  When you resubmit, please ensure that you provide the correct grant numbers for the awards you received for your study in the ‘Funding Information’ section. 4.  Thank you for stating the following in the Competing Interests/Financial Disclosure * (delete as necessary) section: (We have read the journal's policy and Danielle Gallegos has the following competing interest: she is currently supported by a grant from the Queensland Children's Hospital Foundation provided philanthropically by Woolworths. The remaining authors have declared that no competing interests exist.) We note that you received funding from a commercial source: [Name of Company]Please provide an amended Competing Interests Statement that explicitly states this commercial funder, along with any other relevant declarations relating to employment, consultancy, patents, products in development, marketed products, etc.  Within this Competing Interests Statement, please confirm that this does not alter your adherence to all PLOS ONE policies on sharing data and materials by including the following statement: "This does not alter our adherence to PLOS ONE policies on sharing data and materials.” (as detailed online in our guide for authors http://journals.plos.org/plosone/s/competing-interests).  If there are restrictions on sharing of data and/or materials, please state these. Please note that we cannot proceed with consideration of your article until this information has been declared.  Please include your amended Competing Interests Statement within your cover letter. We will change the online submission form on your behalf.

Reviewers' comments:

Reviewer's Responses to Questions

**Comments to the Author**

1. Is the manuscript technically sound, and do the data support the conclusions?

Reviewer #1: Yes

Reviewer #2: Partly

2. Has the statistical analysis been performed appropriately and rigorously? 

Reviewer #1: I Don't Know

Reviewer #2: Yes

3. Have the authors made all data underlying the findings in their manuscript fully available?

Reviewer #1: Yes

Reviewer #2: Yes

4. Is the manuscript presented in an intelligible fashion and written in standard English?

Reviewer #1: Yes

Reviewer #2: Yes

5. Review Comments to the Author

Reviewer #1: This is a well-written paper, crystal clear with a logic flow. As I am not familiar with the statistical methods, i am unable to judge whether the methods and related findings are sound. In any case, the presentation of findings is to the point and reader-friendly.

My main remark regards the way in which the definition of 'water security' as meant in its broadest sense (rows 315 - 319) is interpreted in a very narrow meaning to just water for domestic uses and sanitation - for which WHO/UN have made their own indicators, also for monitoring SDG 6.1. For example, water plays a vital, possibly more important role than just drinking water, or all domestic uses, in food security/nutrition and related physical and mental health, including through irrigation of rice and fisheries/salt intrusion in VMD.

Second, can the authors clarify what 'purchased' water is in table 1? What are the water quality problems with that, if any? Is purchased water only for drinking? What about the much higher volumes needed for other domestic uses?

Minor remarks

row 37: been = between?

row 107: further specify whether urban/rural?

table 2: last column: the Khmer are compared with the Kinh of the northern province. Yet, the Khmer live mainly in the southern province. Hence, wouldn't comparison with SP be more relevant here - keeping more factors constant?

row 299: food = food insecurity?

And a suggestion: for the further exploration of pathways through which water security for domestic uses and food security/ nutrition are linked, see, for example, IFPRI, e.g. Claudia Ringler.

Lastly, some philosophical thoughts from a sociological/institutional perspective, not for this paper, but more general. The SDGs already emphasize the multiple linkages between the SDGs. There is quite some qualitative literature about linkages between water for drinking and other domestic uses, and and health, e.g., infant morbidity (with contestations, though). System thinking is promoted. However, cross-sectoral collaboration is notoriously difficult, as also well recognized in the SDGs. The authors' narrow interpretation of the broad concept of 'water security' is another case in point: even within the water sector, the silos between WASH and other (irrigation, water resource management, pollution, climate etc.) are rigid. Yet, evidence shows that in rural areas, water supply systems designed for domestic uses (for healthy absorption of nutrients) are, in reality, often also used for productive uses (for the intake of nutrients). Certainly for rural people with multi-faceted agrarian livelihoods, it is obvious that they need water for both domestic and often a range of productive uses. Hence, the proposed combined water and food security programs are easier said than done.

Reviewer #2: The research is generally rigorous and well described in the manuscript. However, I do have some concerns about some aspects of the paper as per the below comments:

1. The drinking water treatment variable has three categories: point of use, outside treated and untreated. Yet, the untreated category includes a variety of sources with significantly different safety profiles. Within the WHO-UNICEF JMP system for categorise water sources - boreholes/tubewells are an improved source as when properly designed and maintain protect water for most sources of contamination, and hence treatment is often no required. Whereas, surface water is significantly more likely to be contaminated and is very risky when not treated. Hence, I don't think that current categorisation tells you much about the safety of water being consumed and therefore in its current design that particular variable does not tell us much about water security (which is what you are ultimately interested in). This calls into question the results connected to that particular variable in my mind (for example results on L232-3 and others). There are various other more minor concerns I have with that variable such as putting treated piped supply in with bottle water, and also the implication that POT is the highest outcome when we no of the many flaws in POT in terms of behavioural adherence.

2. The discussion about the conceptualisation of water security is also limited and does not engage sufficiently with the extremely deep literature in this area. For example, the HWISE initiative which has mapped out these key components before in many papers, but even more broadly any literature on monitoring water supply within development programs (including grey literature) can already tell you about the various components you mention. I don't really see how you have developed any sort of novel conceptualisation of water security.

3. Relatedly, the links between WASH and food insecurity are well known and articulated normally from a WASH-nutrition perspective. The links between WASH and nutrition were linked to improving child health with the need to protect against infection as well as ensure nourishment to avoid childhood stunting and to support healthy children. Hence, reflecting back on that trend and the evidence base that underpins it would be help enrich your framing of this project.

4. My final comment is about the "so what" of the paper. I thought overall the study seemed rigorous but I was left wondering why assess these interlinkages when we know (or can at least theorise) that they exist. It is a bit like when studying multidimensional poverty - many things are interlinking and correlating but studying that often doesn't really tell us anything useful. Hence, what does the paper really tell us about these relationships? Does it provide more precision or accuracy in how they relate? Are there particular directions of causality that can identified? What are the implications for governments and others who need to help respond to these problems? I say these comments not to suggest the paper is not valuable, but rather to hopefully help you refine and explain the value to readers. Best of luck.

6. PLOS authors have the option to publish the peer review history of their article (what does this mean?). If published, this will include your full peer review and any attached files.

Reviewer #1: No

Reviewer #2: **Yes: **Paul Hutchings

---

## [Author Response · Author response to Decision Letter 0]

29 Mar 2022

Please note the specific responses to each reviewer is provided in the response to reviewers document appended to this submission. The response to editors is provided in the above statement and in the cover letter

---

## [Editor Report · Decision Letter 1]

7 Apr 2022

Household water and food insecurity negatively impacts self-reported physical and mental health in the Vietnamese Mekong Delta

PONE-D-21-39513R1

Dear Dr. Gallegos,

We’re pleased to inform you that your manuscript has been judged scientifically suitable for publication and will be formally accepted for publication once it meets all outstanding technical requirements.

Kind regards,

Alison Parker

Academic Editor

PLOS ONE
---

## [Editor Report · Acceptance letter]

27 Apr 2022

PONE-D-21-39513R1 

Household water and food insecurity negatively impacts self-reported physical and mental health in the Vietnamese Mekong Delta 

Dear Dr. Gallegos:

I'm pleased to inform you that your manuscript has been deemed suitable for publication in PLOS ONE. Congratulations! Your manuscript is now with our production department. 

Kind regards, 

on behalf of

Dr. Alison Parker 

Academic Editor

PLOS ONE